

# New method to determine black carbon mass size distribution

Weilun Zhao[1], Gang Zhao[2], Ying Li[3,4], Song Guo[2], Nan Ma[5], Lizi Tang[2], Zirui Zhang[2], Chunsheng Zhao[1]

[1]Department of Atmospheric and Oceanic Sciences, School of Physics, Peking University, Beijing 100871, China

[2]State Key Joint Laboratory of Environmental Simulation and Pollution Control, College of Environmental Sciences and Engineering, Peking University, Beijing 100871, China

[3]Department of Ocean Science and Engineering, Southern University of Science and Technology, Shenzhen 518055, China

[4]Southern Marine Science and Engineering Guangdong Laboratory, Guangzhou 511458, China

[5]Institute for Environmental and Climate Research, Jinan University, Guangzhou 511443, China

*Correspondence to*: Chunsheng Zhao (zcs@pku.edu.cn)

**Abstract.** Black carbon (BC) is an important atmospheric component with strong light absorption. Many attempts have been made to measure BC mass size distribution (BCMSD) for its significant impact on climate and public health. Larger-coverage BCMSD, ranging from upper submicron to larger than 1 μm, contributes to substantial proportion of BC absorption. However, current time resolution of larger-coverage BCMSD measurement was limited to 1 day, which was insufficient to characterize variation of larger-coverage BCMSD. In this study, a new method to determine BCMSD was proposed from size-resolved absorption coefficient measured by an aerodynamic aerosol classifier in tandem with an aethalometer. The proposed method could measure larger-coverage BCMSD with time resolution as high as 1 hour and was validated by comparing the measurement results with that measured by a differential mobility analyzer in tandem with a single particle soot photometer (DMA – SP2) for particle size larger than 300 nm during a field measurement in Yangtze River Delta. Bulk BC mass concentration ($m_{BC,bulk}$) by DMA – SP2 was underestimated by 33 % compared to that by this method because of the limited size range of measurement for DMA – SP2. Uncertainty analysis of this method was performed with respect to mass absorption cross-section (MAC), transfer function inversion, number fraction of BC-containing particle and instrumental noise. The results indicated that MAC was the main uncertainty source, leading to $m_{BC,bulk}$ varied from – 20 % to 28 %. With the advanatage of wide size coverage up to 1.5 μm, high time resolution, easy operation and low cost, this method is expected to have wide applications in field measurement for better estimating radiative properties and climate effects of BC.

## 1 Introduction

Atmospheric black carbon (BC) is emitted through incomplete combustion of carbon-based fuels (Bond, 2001), such as biomass burning and fossil fuel combustion (Bond et al., 2004). BC warms atmosphere and is considered the third important light absorber in the climate system after $CO_2$ and $CH_4$ (Bond et al., 2013). BC can induce the respiratory and cardiovascular disease through inhalation (Highwood and Kinnersley, 2006). Plenty of studies have been devoted to BC for its significant impact on the climate and public health.

Bulk BC mass concentration ($m_{BC,bulk}$) is one of the essential parameters for modeling because radiative transfer models calculate



BC absorption from $m_{BC,bulk}$ (Bond et al., 2013). A great deal of research has been dedicated to $m_{BC,bulk}$ measurement for both
model assimilation and environmental monitoring (Castagna et al., 2019;Helin et al., 2018;Ran et al., 2016). A recent study indicated
that the radiative effect of BC was extremely sensitive to its particle size (Matsui et al., 2018). Zhao et al. (2019) further revealed
that the variation of BC mass size distribution (BCMSD), namely size-resolved BC mass concentration ($m_{BC,size-resolved}$), led to
substantial changes in the radiative effect of BC based on field measurement, highlighting the importance of BCMSD on modeling
the radiative effect of BC rather than simply $m_{BC,bulk}$. The size of BC affects the deposition rate of BC to the lung (Highwood and
Kinnersley, 2006), indicating that BCMSD is closely related to health. In the ambient environment, BCMSD is influenced by
emission sources. For instance, BCMSD of fossil fuel combustion differs obviously from that of biomass burning (Schwarz et al.,
2008), implying that BCMSD is one of the characteristics of emission source. The temporal variation of BCMSD can reflect the
atmospheric aging of BC, during which BC undergoes remarkable change in size, accompanied by dramatical variation of BC optical
properties (Zhang et al., 2008). Therefore, reliable measurement and understanding of BCMSD are highly necessary for estimating
the impact of BC on both the earth energy budget and public health (Moosmuller et al., 2009).
Quite a few efforts was made to measure BCMSD. The single-particle soot photometer (SP2) was developed using laser-induced
incandescence (Schwarz et al., 2006), which measured BCMSD on a single-particle level. The soot particle aerosol mass
spectrometer (SP-AMS) combined laser-induced incandescence and mass spectroscopy, which could determine not only BCMSD
but also the chemical composition of BC-containing particles (Onasch et al., 2012). The SP2 and SP-AMS techniques were
characterized by high time resolution and high accuracy, but high cost and complicated maintenance, as a result, not widely applied
for routine measurement. A more convenient solution was required for wider and better characterization of BCMSD in different
regions and emission sources. BCMSD could be sampled by multi-stage cascade impactor (Viidanoja et al., 2002) combined with
off-line analysis, such as thermo/optical organic carbon/elemental carbon method (Chow et al., 2001). BCMSD sampled by multi-
stage cascade impactor had wide size coverage, but low time resolution, usually from 24 hours (Soto-Garcia et al., 2011) to 48 hours
(Guo, 2015), which was too low to resolve aging of BC. Differential mobility analyzer (DMA) in tandem with filter-based instrument
(Hansen et al., 1984), for instance, micro-aethalometer (MA) (Ning et al., 2013) and particle soot absorption photometer (PSAP)
(Tunved et al., 2021), was used to determine BCMSD with higher time resolution up to 2 hours (Zhao et al., 2021). However, the
multiple-charge correction and low flow rate of DMA imposed restrictions on the accuracy of the measured BCMSD. The measured
size range of DMA was limited to less than about 700 nm, resulting in incomplete measured BCMSD. Current measurement of
larger-coverage BCMSD ranging from upper submicron to larger than 1 μm was limited in time resolution. Characteristics of larger-
coverage BCMSD during atmospheric aging was still unclear. Therefore, it was imperative to measure larger-coverage BCMSD
with higher time resolution.
In this study, a new method to determine BCMSD was proposed using size-resolved absorption coefficient ($\sigma_{ab,size-resolved}$)
measured by an aerodynamic aerosol classifier (AAC, Cambustion, UK, Tavakoli and Olfert (2013)) in tandem with an aethalometer
(model AE33, Magee, USA, Drinovec et al. (2015)), combined with size-resolved number concentration ($N_{size-resolved}$),



simultaneously measured by scanning mobility particle sizer (SMPS, TSI, USA) and an aerodynamic particle sizer (APS, TSI, USA).
The proposed method for determining BCMSD overcame the disadvantages and weighed the advantages of the above-mentioned
methods, characterized by high cost-effectiveness, easy and convenient maintenance, high time resolution to 1 hour, and wide size
range to up to 1.5 μm. The proposed method was validated in a field measurement in the Yangtze River Delta and the uncertainty
study was carried out based on the measured data.
**2 Experimental setup**
**2.1 Instrumental setup**
Figure 1 illustrated the instrumental setup developed in this study, which could be split into two parts, namely the measurement
setup and the validation setup. Ambient aerosol particles were dried to relative humidity (RH) less than 30 % beforehand. For the
measurement setup, AAC in tandem with AE33 (AAC – AE33) measured $\sigma_{\mathrm{ab,size-resolved}}$ at a flow rate of 3 L min⁻¹. Since BCMSD
of larger size coverage was mainly focused in this study, AAC was set to scan 12 logarithmically spaced particle sizes ($D_{\mathrm{p}}$) from
200 nm to 1500 nm. Each size was scanned for 5 minutes and 1 hour was required for a complete cycle. It should be noted that $D_{\mathrm{p}}$
was aerodynamic size in this study. Mobility size related to DMA was converted to aerodynamic size assuming an effective density
of 1.3 g cm⁻³ (Zhao et al., 2019;DeCarlo et al., 2005). AE33 measured absorption coefficient ($\sigma_{\mathrm{ab}}$) at 7 wavelengths from 370 nm
to 950 nm (Drinovec et al., 2015), at which 880 nm was adopted in this study because BC dominated particle absorption at 880 nm
(Ramachandran and Rajesh, 2007). SMPS and APS measured $N_{\mathrm{size-resolved}}$ for $D_{\mathrm{p}}$ less than and greater than about 800 nm at a
flow rate of 0.3 L min⁻¹ and 5 L min⁻¹, respectively.
For validation system, the tandem array of Neutralizer, DMA and SP2 (DMA – SP2) measured BCMSD (BCMSD$_{\mathrm{DMA-SP2}}$) at a
flow rate of 0.12 L min⁻¹ for purpose of comparing with BCMSD determined by the proposed method (BCMSD$_{\mathrm{AAC-AE33}}$). The lower
detection limit of $D_{\mathrm{p}}$ for DMA – SP2 was about 200 nm (Zhao et al., 2020a). Therefore, lower $D_{\mathrm{p}}$ limit for both BCMSD$_{\mathrm{DMA-SP2}}$
and BCMSD$_{\mathrm{AAC-AE33}}$ was set as 200 nm in this study. Another AE33 measured bulk absorption coefficient ($\sigma_{\mathrm{ab,bulk}}$) simultaneously
at a flow rate of 2 L min⁻¹ for closure study with $\sigma_{\mathrm{ab,size-resolved}}$.
**2.2 Aethalometer model AE33**
The principle of obtaining $\sigma_{\mathrm{ab}}$ was well developed for aethalometer (Hansen et al., 1984) and described here in brief. Ambient
aerosol particles were drawn into an aethalometer at a flow rate $F$ and collected on an area $S$ of a filter. A light source illuminated
the filter. The transmitted light signal was denoted as $I$ ($I_0$) for the light which passed through the particle-laden (particle-free) part
of the filter. Light attenuation was defined as
$$\mathrm{ATN} = -100 \cdot \ln\left(\frac{I}{I_0}\right). \tag{1}$$
ATN increased with decreasing $I$ as aerosol particles were loaded on the filter continuously. Therefore, ATN reflected aerosol
loading on the filter. If ATN increased $\Delta\mathrm{ATN}$ during time interval $\Delta t$, then attenuation coefficient was defined as
$$\sigma_{\mathrm{ATN}} = \frac{S}{100 \cdot F} \cdot \frac{\Delta\mathrm{ATN}}{\Delta t}. \tag{2}$$





The light attenuation was caused by not only particle absorption, but also scattering by particle and filter matrix. A scattering
parameter $C$ was introduced to extract $\sigma_{ab}$ from $\sigma_{ATN}$:
$\sigma_{ab} = \frac{\sigma_{ATN}}{C}$,                                                                                                                                                (3)
where $C$ was set as 2.9 (Zhao et al., 2020b) in this study. Nonlinearity, termed loading effect, became more and more significant
with increasing aerosol loading, namely for the same increase in aerosol loading, the corresponding increase in ATN was smaller
for heavier aerosol loading. The "dual-spot" technique (Drinovec et al., 2015) was proposed to correct the loading effect for AE33
and was used in this study. The $\sigma_{ab}$ measured by AE33 at given a particle size selected by AAC was termed binned $\sigma_{ab}$ ($\sigma_{ab,binned}$)
in this study to differentiate from $\sigma_{ab,bulk}$.

**2.3 Aerodynamic aerosol classifier**

The principle of AAC was illustrated detailedly by Tavakoli and Olfert (2013) and was introduced here concisely. The setup of
AAC could be simplified to two coaxial cylinders, where the inner radius, outer radius and length was denoted as $r_i$, $r_o$ and $L$.
Polydisperse particles flowed into the space between the inner cylinder and outer cylinder from one end of the inner cylinder at a
flow rate of $Q_{sample}$. Meawhile, Particle free sheath flow streamed in the space between the inner cylinder and outer cylinder in the
direction of the axis of the coaxial cylinders at a flow rate of $Q_{sheath}$. The sheath flow carried the particles along the coaxial cylinders.
At the same time, the two coaxial cylinders rotated with respect to their axis at a rotational speed of $\omega$. Therefore, the particles was
migrated outwards across the sheath flow by the centrifugal force acting on them. Relaxation time ($\tau$) was defined as
$\tau = \frac{C_c(D_p)\rho_0 D_p^2}{18\mu}$,                                                                                                                                        (4)
where $C_c(D_p)$ was the cunningham slip correction factor as a function of $D_p$ (Kim et al., 2005), $\rho_0 = 1\ \text{g cm}^{-3}$ was the reference
density and $\mu$ was the gas viscosity. It could be seen that $\tau$ was directly related to $D_p$. Dynamic analysis proved that only particles
with certain relaxation time $\tau$, and hence certain $D_p$, could migrate to another end of outer cylinder and emerge as monodisperse
flow. $\tau$ was related to parameters of AAC by
$\tau = \frac{2Q_{sh}}{\pi\omega^2(r_i+r_o)^2 L}$.                                                                                                                                   (5)
Therefore, by changing $\omega$ and $Q_{sh}$, monodisperse particles of desired $D_p$ could be selected by AAC. Unlike DMA, particles did
not need to be charged before entering AAC. Consequently, the transmission efficiency ($\lambda_\Omega$) of AAC about 4 times higher than that
of DMA and multi-charge correction was not required for data measured by the instrument downstream AAC (Johnson et al., 2018).

**2.4 Method**

**2.4.1 Deriving size-resolved absorption coefficient from binned absorption coefficient**

Tavakoli and Olfert (2013) formulated the ideal theoretical model for AAC transfer function inversion, which was adopted by this
study to derive $\sigma_{ab,size-resolved}$ from $\sigma_{ab,binned}$ and presented here in short. The $\sigma_{ab,size-resolved}$ was given by
$\sigma_{ab,size-resolved} = \frac{d\sigma_{ab}}{dlogD_p} = \frac{\ln(10)}{\frac{dlogD_p}{dlog\tau}\cdot\beta^*}\cdot\sigma_{ab,binned}$,                                              (6)



where $\beta^*$ was related to the ratio of $Q_{\text{sheath}}$ to $Q_{\text{sample}}$, $R_{\text{t}} = \frac{Q_{\text{sheath}}}{Q_{\text{sample}}} = \frac{1}{\beta}$, through
$$\beta^* = \left(1 + \frac{1}{\beta}\right)\ln(1 + \beta) - \left(1 - \frac{1}{\beta}\right)\ln(1 - \beta). \tag{7}$$
Johnson et al. (2018) corrected the ideal inversion formula (6) to take particle loss and spectral broadening into account by replacing
$\beta^*$ with a nonideal $\beta^*_{\text{NI}}$:
$$\beta^*_{\text{NI}} = \lambda_\Omega \mu_\Omega \left[\ln\left(\frac{1 - \beta/\mu_\Omega}{1 + \beta/\mu_\Omega}\right) + \frac{\mu_\Omega}{\beta}\ln\left(1 - \left(\frac{\mu_\Omega}{\beta}\right)^2\right)\right], \tag{8}$$
where $\mu_\Omega$ was the spectral broadening factor. Both $\lambda_\Omega$ and $\mu_\Omega$ were dependent on $D_{\text{p}}$ as well as flow, and discussed in detail in
Sect. 4.2.

**2.4.2 Deriving black carbon mass size distribution from size-resolved absorption coefficient**

$\sigma_{\text{ab,size-resolved}}$ could be converted to $\text{BCMSD}_{\text{AAC-AE33}}$ through mass absorption cross-section (MAC) (Bond and Bergstrom,
2006), which was assumed a fixed value of 7.77 m²/g for AE33 (Drinovec et al., 2015). However, both field measurement (Bond
and Bergstrom, 2006) and model study (Zhao et al., 2021) manifested that MAC varied significantly. A fixed MAC led to uncertainty
in derived $\text{BCMSD}_{\text{AAC-AE33}}$. In this study, MAC was variable based on method proposed by Zhao et al. (2021) for more accurate
$\text{BCMSD}_{\text{AAC-AE33}}$ estimation. Briefly, a 2-dimensional lookup table of MAC with respect to $D_{\text{p}}$ and BC core diameter ($D_{\text{c}}$) was
simulated ($\text{MAC}(D_{\text{p}}, D_{\text{c}})$) by Mie theory assuming a core-shell geometry. At a given size bin centered at $D_{\text{p}}$, the corresponding $\sigma_{\text{ab}}$
and number concentration ($N$) at the size bin, denoted as $\sigma_{\text{ab}}(D_{\text{p}})$ and $N(D_{\text{p}})$, could be derived by $\sigma_{\text{ab,size-resolved}}$ and
$N_{\text{size-resolved}}$:
$$\sigma_{\text{ab}}(D_{\text{p}}) = \sigma_{\text{ab,size-resolved}}(D_{\text{p}}) \cdot \Delta \log D_{\text{p}}, \tag{9}$$
$$N(D_{\text{p}}) = N_{\text{size-resolved}}(D_{\text{p}}) \cdot \Delta \log D_{\text{p}}, \tag{10}$$
where $\Delta \log D_{\text{p}}$ was the logarithmic bin width of the size bin. The number concentration of BC-containing particle $N_{\text{BC}}(D_{\text{p}})$ at
the size bin was determined by
$$N_{\text{BC}}(D_{\text{p}}) = N(D_{\text{p}}) \cdot f_{\text{BC}}, \tag{11}$$
where $f_{\text{BC}}$ was the number fraction of BC-containing particle, which was assumed as a fixed value in this study and the uncertainty
of the fixed-$f_{\text{BC}}$ assumption was discussed in Sect. 4.3. An optimal $D_{\text{c}}$ was found so that calculated $\sigma_{\text{ab}}$ at the size bin, denoted
as $\sigma_{\text{ab,calc}}(D_{\text{p}})$, matched $\sigma_{\text{ab}}(D_{\text{p}})$:
$$\sigma_{\text{ab,calc}}(D_{\text{p}}, D_{\text{c}}) = \rho_{\text{BC}} \frac{\pi}{6} D_{\text{c}}^3 \cdot \text{MAC}(D_{\text{p}}, D_{\text{c}}) \cdot N_{\text{BC}}(D_{\text{p}}) = \sigma_{\text{ab}}(D_{\text{p}}), \tag{12}$$
where $\rho_{\text{BC}}$ was the density of BC, and set as 1.8 g cm⁻³ (Bond and Bergstrom, 2006), consistent with the $\rho_{\text{BC}}$ assumption when
deriving $\text{BCMSD}_{\text{DMA-SP2}}$. BC mass concentration ($m_{\text{BC}}$) at the size bin, denoted as $m_{\text{BC}}(D_{\text{p}})$, could be determined by
$$m_{\text{BC}}(D_{\text{p}}) = \frac{\sigma_{\text{ab}}(D_{\text{p}})}{\text{MAC}(D_{\text{p}}, D_{\text{c}})}, \tag{13}$$
then the BCMSD at $D_{\text{p}}$, denoted by $\text{BCMSD}(D_{\text{p}})$, could be determined by
$$\text{BCMSD}(D_{\text{p}}) = \frac{m_{\text{BC}}(D_{\text{p}})}{\Delta \log D_{\text{p}}}. \tag{14}$$





**2.5 Field measurement**

The proposed method was applied to a field measurement in Changzhou, Jiangsu Province, China (119°36′E, 31°43′N), located in the Yangtze River Delta, from May 17th to June 3rd in 2021. Changzhou was between two megacities, about 187 km to the northwest of Shanghai and about 82 km to the southeast of Nanjing, as shown in Fig. S1a. The area between the Shanghai and Nanjing underwent serious environmental pollution in the past few decades with the development of industrialization and urbanization. As shown in Fig. S1b, the pollution condition of Changzhou was overall representative of the regional background pollution in the Yangtze River Delta.

**3 Results and discussion**

Figure 2 presented the time series of the measurement results. There were 4 pollution episodes during the measurement period: from May 18th to May 19th, from May 21st to May 22nd, from May 24th to May 26th, and from May 29th to May 31st. Both $\mathrm{BCMSD_{AAC-AE33}}$ (Fig. 2b) and $\mathrm{BCMSD_{DMA-SP2}}$ (Fig. 2c) recorded the 4 pollution episodes simultaneously with higher BCMSD values than clean episodes. $\mathrm{BCMSD_{AAC-AE33}}$ and $\mathrm{BCMSD_{DMA-SP2}}$ were both integrated from 200 nm to 720 nm, which was the $D_\mathrm{p}$ range of measurement for DMA − SP2, and the results were denoted as $m_\mathrm{BC,bulk,AAC-AE33,part}$ and $m_\mathrm{BC,bulk,DMA-SP2}$, respectively. As shown in Fig. 2a, $m_\mathrm{BC,bulk,AAC-AE33,part}$ compared well with $m_\mathrm{BC,bulk,DMA-SP2}$ with determination coefficient ($R^2$), slope ($b_1$), and intercept ($b_0$) of 0.8 (accurate to one decimal place), 1.0 and 0.1 μg m⁻³ (Fig. S2), indicating the proposed method was capable of capturing the variation of $m_\mathrm{BC,bulk}$. The mean $m_\mathrm{BC,bulk,AAC-AE33,part}$ and $m_\mathrm{BC,bulk,DMA-SP2}$ were 0.7 ± 0.4 μg m⁻³ and 0.6 ± 0.3 μg m⁻³, indicating $m_\mathrm{BC,bulk,AAC-AE33,part}$ was overall 0.1 μg m⁻³ higher than $m_\mathrm{BC,bulk,DMA-SP2}$, consistent with $b_1$ of 0.1 μg m⁻³. The reason for overall discrepancy of 0.1 μg m⁻³ in $m_\mathrm{BC,bulk}$ might be that DMA − SP2 could not detect BC with $D_\mathrm{c}$ less than about 100 nm (Zhao et al., 2020a), resulting in an underestimated $m_\mathrm{BC,bulk,DMA-SP2}$. $\mathrm{BCMSD_{AAC-AE33}}$ was also integrated from 200 nm to 1500 nm, which was the $D_\mathrm{p}$ range of measurement for DMA − SP2, and the result was denoted as $m_\mathrm{BC,bulk,AAC-AE33}$. $R^2$ decreased to 0.7, $b_1$ and $b_0$ increased to 1.2 and 0.2 μg m⁻³ between $m_\mathrm{BC,bulk,AAC-AE33}$ and $m_\mathrm{BC,bulk,DMA-SP2}$. The mean $m_\mathrm{BC,bulk,AAC-AE33}$ was 0.9 ± 0.5 μg m⁻³, ~ 0.2 μg m⁻³ higher than $m_\mathrm{BC,bulk,AAC-AE33,part}$, indicating that DMA − SP2 overall underestimated $m_\mathrm{BC,bulk}$ for ~ 0.2 μg m⁻³ (~ 33 %) in this field measurement considering that DMA − SP2 could not measure BCMSD larger than about 720 nm. The decrease in $R^2$ indicated BCMSD of larger $D_\mathrm{p}$ contained more sophisticated structure. Therefore, it was highly necessary to measure BCMSD with wider $D_\mathrm{p}$ range for better estimation of $m_\mathrm{BC,bulk}$.

Figure 3 exhibited the mean $\mathrm{BCMSD_{AAC-AE33}}$ ($\overline{\mathrm{BCMSD}}_\mathrm{AAC-AE33}$) and mean $\mathrm{BCMSD_{DMA-SP2}}$ ($\overline{\mathrm{BCMSD}}_\mathrm{DMA-SP2}$) during the field measurement. It could be seen that when $D_\mathrm{p}$ was less than about 300 nm, $\overline{\mathrm{BCMSD}}_\mathrm{AAC-AE33}$ was higher than $\overline{\mathrm{BCMSD}}_\mathrm{DMA-SP2}$. The higher $\overline{\mathrm{BCMSD}}_\mathrm{AAC-AE33}$ may be due to particle diffusion at small $D_\mathrm{p}$ which was not well corrected by (7) and underestimated MAC. When $D_\mathrm{p}$ was greater than 300 nm and less than about 700 nm, $\overline{\mathrm{BCMSD}}_\mathrm{AAC-AE33}$ was overall consistent with $\overline{\mathrm{BCMSD}}_\mathrm{DMA-SP2}$. When $D_\mathrm{p}$ was larger than 700 nm, where DMA − SP2 could not measure, $\overline{\mathrm{BCMSD}}_\mathrm{AAC-AE33}$ decreased with increasing $D_\mathrm{p}$ when $D_\mathrm{p}$ less than about 870 nm, and increased with increasing $D_\mathrm{p}$ when $D_\mathrm{p}$ was larger than 870 nm. In the study





by Yu et al. (2010), three modes of BCMSD were identified: the mode peaked at about 400 nm, 1000 nm and 5000 nm, which were
termed as condensation mode, droplet mode and coarse mode, respectively. Following the nomenclature proposed by Yu et al. (2010),
the increasing (decreasing) $\overline{\text{BCMSD}}_{\text{AAC-AE33}}$ with increasing $D_{\text{p}}$ for $D_{\text{p}}$ larger (less) than 870 nm signified the lower (upper) end
of droplet mode (condensation mode). However, $\overline{\text{BCMSD}}_{\text{DMA-SP2}}$ only identified condensation mode, which clearly highlighted
the necessity to measure BCMSD of wider $D_{\text{p}}$ range for better characterization of BCMSD. Both the proposed method and DMA
– SP2 showed that the temporal variation of BCMSD, expressed as standard deviation (std) of BCMSD in Fig. 3, was as large as
$\overline{\text{BCMSD}}$, reflecting the complex mechanism in the variability of BCMSD.
**4 Uncertainty analysis**
Uncertainty analysis was performed with respect to the MAC lookup table, transfer function inversion from $\sigma_{\text{ab,binned}}$ to
$\sigma_{\text{ab,size-resolved}}$, $f_{\text{BC}}$ and instrumental noise, respectively. The resulting uncertainty to $\text{BCMSD}_{\text{AAC-AE33}}$ was illustrated in Fig. 4
and to $m_{\text{BC,bulk}}$ was shown in table 1. It could be seen from Fig. 4 that the boundary between condensation mode and droplet mode
was distinct no matter how the uncertainty source disturbed $\text{BCMSD}_{\text{AAC-AE33}}$.
**4.1 Uncertainty from masss absorption cross-section**
According to Zhao et al. (2021), the variation in refractive index (RI) dominated the uncertainty of the MAC lookup table.
Therefore, the uncertainty from the MAC lookup table was analyzed with respect to RI. The real part of RI (Re[RI]) was reported
to vary from 1.5 to 2.0 in general (Liu et al., 2018) and the imaginary part of RI (Im[RI]) ranged from 0.5 to 1.1 commonly (Bond
and Bergstrom, 2006). Hence, Re[RI] (Im[RI]) was changed from 1.5 (0.5) to 2.0 (1.1) with step increase of 0.01, the resulting mean
MAC ($\overline{\text{MAC}}$) was the MAC lookup table used in this study (Fig. S3a) and the resulting std divided by the $\overline{\text{MAC}}$ was considered as
the uncertainty of the MAC lookup table (Fig. S3b). As shown in Fig. S2b, when $D_{\text{c}}$ was larger than about 400 nm, the uncertainty
was less than 10% and influenced by both $D_{\text{p}}$ and $D_{\text{c}}$. However, when $D_{\text{c}}$ was less than 400 nm, the uncertainty increased rapidly
with decreasing $D_{\text{c}}$ and was dominated by $D_{\text{c}}$. The uncertainty increased to larger than 23 % when $D_{\text{c}}$ was less than about 100
nm. When $D_{\text{p}}$ was less than about 300 nm, the uncertainty varied from 14% to 24% with a mean value of 22 %, indicating large
uncertainty in $\text{BCMSD}_{\text{AAC-AE33}}$ for $D_{\text{p}}$ less than 300 nm.
The MAC lookup table was replaced with original $\overline{\text{MAC}}$ minus its std (-stdMAC) and plus its std (+stdMAC). The resulting
MAC lookup tables were taken into the method proposed by Zhao et al. (2021), and applied to the entire measurement period to
study the influence of MAC variation on the $\text{BCMSD}_{\text{AAC-AE33}}$. $\overline{\text{BCMSD}}_{\text{AAC-AE33}}$ and mean $\text{BCMSD}_{\text{AAC-AE33}}$ of $\pm$stdMAC were
shown in Fig. 4a. The uncertainty increased with decreasing $D_{\text{p}}$, and reached to maximum when $D_{\text{p}}$ was less than 300 nm,
indicating the $\text{BCMSD}_{\text{AAC-AE33}}$ for $D_{\text{p}}$ less than 300 nm might potentially have nonnegligible uncertainty. The uncertainty in bulk
$m_{\text{BC}}$ was from – 20 % (+stdMAC) to + 28 % (-stdMAC), which was the largest among the four uncertainty sources, as shown in
Table 1.
**4.2 Uncertainty from the transfer function inversion**
As stated in Sect. 2.4.1, correction for particle loss and spectral broadening was required when $\sigma_{\text{ab,binned}}$ was converted to





$\sigma_{ab,size-resolved}$. $\lambda_{\Omega}$ was defined to correct particle loss where $\lambda_{\Omega} = 0$ ($\lambda_{\Omega} = 1$) stood for total (no) particle loss. The relationships
between $\lambda_{\Omega}$ and $D_p$ as well as $Q_{sample}$, as shown in Fig. S4a, were well developed (Karlsson and Martinsson, 2003) and
consistent with experimental data of AAC (Johnson et al., 2018). $Q_{sample}$ used in this study was 3.0 L min$^{-1}$. $Q_{sample}$ was changed
from $-30\%$ (2.1 L min$^{-1}$) to $+30\%$ (3.9 L min$^{-1}$), and the resulting $\lambda_{\Omega}$ was used to study the fluctuation of $Q_{sample}$ on $\lambda_{\Omega}$. As
shown in Fig. S4a, the variation of $\lambda_{\Omega}$ was less than 0.5 %, which was negligible.
Spectral broadening was caused by small-size particle diffusion as well as fluctuation of sheath flow and described by $\mu_{\Omega}$ where
$\mu_{\Omega} < 1$ ($\mu_{\Omega} = 1$) was for (no) broadening. $\mu_{\Omega}$ was parameterized because of the complicated fluid dynamics and its interaction
with particles. Johnson et al. (2018) parameterized $\mu_{\Omega}$ based on $R_t = 10$ (Fig. S4b). However, $R_t$ was about 2.5 in this study,
which might lead to uncertainty. $\mu_{\Omega}$ was varied from $-23\%$ to $+30\%$ to study the impact of $\mu_{\Omega}$ on $BCMSD_{AAC-AE33}$. The reason
for the lower limit of $\mu_{\Omega}$ set as $-23\%$ rather than $-30\%$ was that $BCMSD_{AAC-AE33}$ was negative when $\mu_{\Omega}$ was less than $-23\%$.
The resulting influence on the $BCMSD_{AAC-AE33}$ was shown in Fig. 4b. The uncertainty of $\mu_{\Omega}$ did not exhibit a significant size
dependence. Lower $\mu_{\Omega}$ led to lower $BCMSD_{AAC-AE33}$, and vice versa. It should be noted that the uncertainty in the $m_{BC,bulk}$ was
from $-1\%$ ($-23\%$ of $\mu_{\Omega}$) to $+21\%$ ($+30\%$ of $\mu_{\Omega}$), implying systematic overestimation of $m_{BC,bulk}$. Therefore, $\mu_{\Omega}$ was replaced
with $-23\%$ of its original value in this work to offset the bias due to the incomplete $\mu_{\Omega}$ parameterization.
$\sigma_{ab,size-resolved}$ (Fig. S5b) was integrated and the result, denoted as $\sigma_{ab,bulk,calc}$, was compared to $\sigma_{ab,bulk}$. As shown in Fig.
S5a, $\sigma_{ab,bulk,calc}$ was consistent with $\sigma_{ab,bulk}$. $R^2$, $b_1$ and $b_0$ between $\sigma_{ab,bulk,calc}$ and $\sigma_{ab,bulk}$ was 0.9, 1.1, and 0.5 Mm$^{-1}$
(Fig. S6), respectively, which validated conversion from $\sigma_{ab,binned}$ to $\sigma_{ab,size-resolved}$.
**4.3 Uncertainty from number fraction of BC-containing particle**
BC-containing aerosol particles had complicated mixing states. Even for internally-mixed particles with same $D_p$, the internal
BC cores might have different $D_c$, which could not be resolved by AAC – AE33. Field measurement (Liu et al., 2010) revealed that
$f_{BC}$ varied with time, $D_c$ and $D_p$. This complexity was simplified to a parameterized fixed value of $f_{BC}$ in this study. A fixed $f_{BC}$
implied that only one $D_c$ value corresponded to a given $D_p$ and the size-resolved number concentration of BC-containing particle
was determined by $N_{size-resolved}$ times $f_{BC}$. Zhao et al. (2021) used $f_{BC}$ value of 0.17 based on SP2 measurement. However,
SP2-derived $f_{BC}$ could not represent the bulk $f_{BC}$ over the whole particle size spectrum due to the detection limit of SP2. And
different regions might have different $f_{BC}$. In this study, $f_{BC}$ was varied and the resulting $m_{BC,bulk,AAC-AE33,part}$ was compared
with $m_{BC,bulk,DMA-SP2}$. $f_{BC}$ was set as 0.35 in this study because $b_1$ was 1.0 between $m_{BC,bulk,AAC-AE33,part}$ and
$m_{BC,bulk,DMA-SP2}$ when $f_{BC}$ was 0.35.
$f_{BC}$ was varied from 0.25 ($-30\%$ of 0.35) to 0.46 ($+30\%$ of 0.35) to analyze the influence of $f_{BC}$ on the $BCMSD_{AAC-AE33}$, as
shown in Fig. 4c. $BCMSD_{AAC-AE33}$ was more influenced around 870 nm. The variation of $f_{BC}$ led to uncertainty of $\pm3\%$ in
$m_{BC,bulk}$, indicating that simplification of $f_{BC}$ was a minor uncertainty source compared to MAC and transfer function inversion.
**4.4 Uncertainty from instrumental noise**
The instrumental noise stemmed from the fluctuation of the light source and flow of AE33, which was reflected as fluctuation in



$I$, $I_0$ and $F$, further leading to the fluctuation in ATN, $\sigma_{ATN}$ and $\sigma_{ab}$. The instrumental noise was defined as the std of $\sigma_{ab,binned}$
and was shown in Fig. S7b. It could be seen that the instrumental noise did not exhibit significant dependence on $D_p$. Comparing
Fig. S7a and Fig. S7b, the instrumental noise was irrelevant to the value of $\sigma_{ab,binned}$. Figure S7c illustrated that the instrumental
noise was also not correlated to $\sigma_{ab,bulk}$, implying that the instrumental noise was not dependent on the pollution level.
The std of instrumental noise was added to (subtracted from) $\sigma_{ab,binned}$ and the derived $BCMSD_{AAC-AE33}$ was used to study the
influence of instrumental noise on $BCMSD_{AAC-AE33}$. The mean result was shown in Fig. 4d. $BCMSD_{AAC-AE33}$ larger than 1000
nm was more influenced by the instrumental noise than $BCMSD_{AAC-AE33}$ smaller than 500 nm. $\sigma_{ab,binned}$ larger than 1000 nm
was relatively small (about 0.3 Mm$^{-1}$) compared to $\sigma_{ab,binned}$ less than 870 nm (about 0.9 Mm$^{-1}$). The mean instrumental noise was
0.1 Mm$^{-1}$ and exhibited no significant dependence on $D_p$. Therefore, $\sigma_{ab,binned}$ larger than 1000 nm was more affected by the
instrumental noise, resulting in distinct variation in $BCMSD_{AAC-AE33}$. Since the mass fraction of $m_{BC,bulk}$ was not dominated by
BCMSD larger than 1000 nm in this study, the resulting uncertainty in $m_{BC,bulk}$ was not obvious, which varied from – 2 % to –
1 %, also minor compared to MAC and transfer function inversion.
**5 Conclusions**
Knowledge of bulk black carbon (BC) characteristics, such as bulk BC mass concentration ($m_{BC,bulk}$), was very limited for deeper
understanding the influence of BC on radiation and health. BC mass size distribution (BCMSD) was one of the BC microphysical
characteristics that could indicate emission source, reflect atmospheric aging and effectively reduce uncertainty related to BC
radiative effect. However, current BCMSD measurement ranging from upper micron to larger than 1 μm was insufficient in time
resolution and sophisiticated for routine measurement. In this study, a new method to determine BCMSD was proposed characterized
by wide size range of measurement up to 1.5 μm, high time resolution up to 1 hour and convenience for extensive measurement.
The BCMSD was retrieved by size-resolved absorption coefficient ($\sigma_{ab,size-resolved}$) measured by an aerodynamic aerosol classifer
in tandem with an aethaelometer model AE33 (AAC – AE33), denoted as $BCMSD_{AAC-AE33}$. Size-resolved number concentration
($N_{size-resolved}$) was measured concurrently by scanning mobility particle sizer (SMPS) and an aerodynamic particle sizer (APS) to
model the influence of size on mass absorption cross-section (MAC).
The proposed method was applied to a field measurement in Yangtze River Delta and validated by comparing the BCMSD with
that measured by an differential mobility analyzer in tandem with a single-particle soot photometer (DMA – SP2), denoted as
$BCMSD_{DMA-SP2}$. The results showed that for particle diameter ($D_p$) less than 300 nm, $BCMSD_{AAC-AE33}$ was higher than
$BCMSD_{DMA-SP2}$, which might be caused by underestimated MAC by the method proposed by Zhao et al. (2021) or incomplete
parameterization of spectral broadening of AAC. $BCMSD_{AAC-AE33}$ was consistent with $BCMSD_{DMA-SP2}$ for $D_p$ larger than 300
nm. $m_{BC,bulk}$ integrated over the size range that AAC – AE33 and DMA – SP2 both measured, denoted as $m_{BC,bulk,AAC-AE33,part}$
and $m_{BC,bulk,DMA-SP2}$, compared well with each other with determination coefficient ($R^2$), slope ($b_1$), and intercept ($b_0$) of 0.8, 1.0
and 0.1 μg m$^{-3}$, respectively. However, DMA – SP2 could not measure $D_p$ larger than 700 nm, leading to 0.2 μg m$^{-3}$ (33 %)
underestimation of $m_{BC,bulk}$, highlighting the necessity to measure BCMSD with a wider size range.





Uncertainty analysis was performed with respect to MAC, transfer function inversion, number fraction of BC-containing ($f_{BC}$)
and instrumental noise. MAC was the largest uncertainty source, leading to significant uncertainty for $D_p$ less than 300 nm and
about 24% uncertainty in $m_{BC,bulk}$. Transfer function inversion was the second largest uncertainty source, which was induced by
incomplete parameterization of spectral broadening. The uncertainty in transfer function inversion led to systematic overestimation
of $m_{BC,bulk}$, which was corrected in this study. Both $f_{BC}$ and instrumental noise were minor uncertainty sources and one order of
magnitude less than MAC and transfer function inversion. $f_{BC}$ was simplification of complicated BC mixing states, leading to
relatively big uncertainty in BCMSD at 870 nm, around the boundary between condensation mode and droplet mode. The BCMSD
for $D_p$ larger than 1000 nm was more sensitive to instrumental noise.
This study proposed a new method to determine BCMSD based on widespread filter-based measurement. The proposed method
was validated by well-designed field measurement and thorough uncertainty analysis, highlighting the necessity to measure BCMSD
with a wider size coverage for a more complete characterization of BCMSD. The new method provided a high-time-resolution,
wide-size-coverage, convenient and cost-effective solution for BCMSD measurement. Hence, the proposed method had great
potential for widespread BCMSD measurement and was expected to promote the research of BC radiative effect, source
apportionment and atmospheric aging.
**Data availability**
The measurement data involved in this study are available upon request to the authors.
**Author contributions**
CZ determined the main goal of this study. WZ and GZ designed the methods. WZ carried them out and prepared the paper with
contributions from all co-authors.
**Competing interests**
The authors declare that they have no conflict of interest.

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

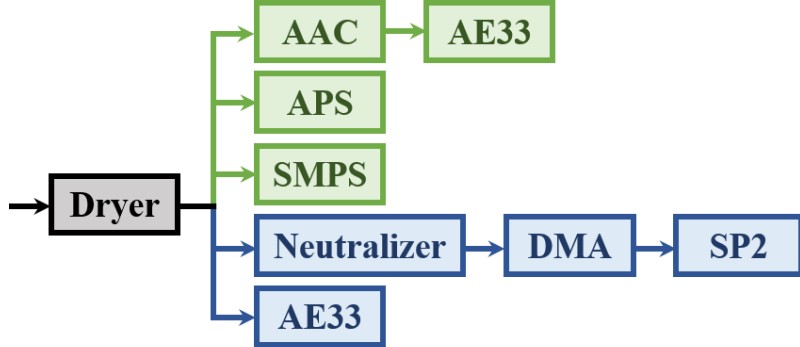


**Figure 1: Schematic diagram of the measurement (green) and the validation (blue) setup.**

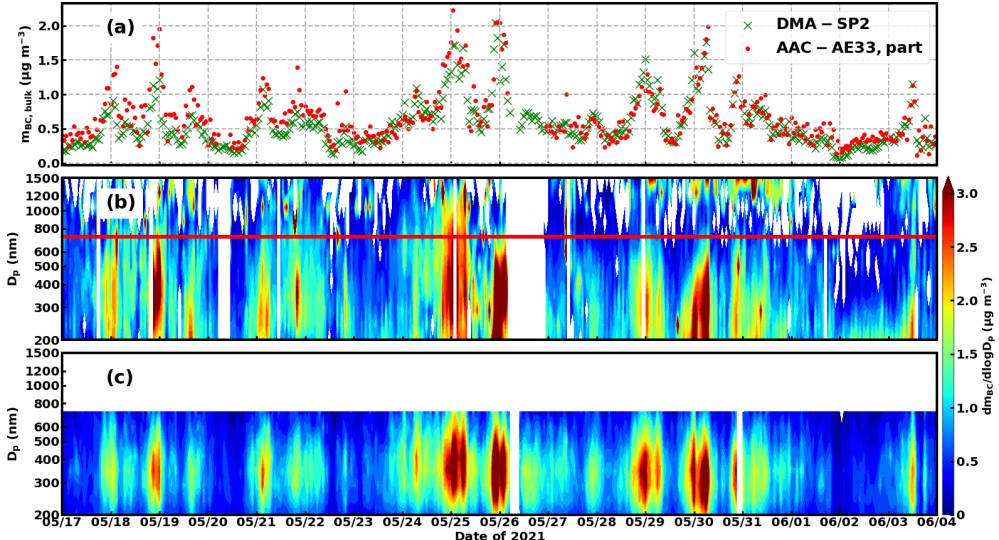


**Figure 2: Time series of (a)** $m_{BC,bulk}$ **from BCMSD integrated from 200 nm to 720 nm determined by the proposed method**

**(red dot, denoted as AAC – AE33, part) and DMA – SP2 (green cross, denoted as DMA – SP2), BCMSD determined by (b)**

**the proposed method and (c) DMA – SP2. The red line in (b) marked particle size of 720 nm.**


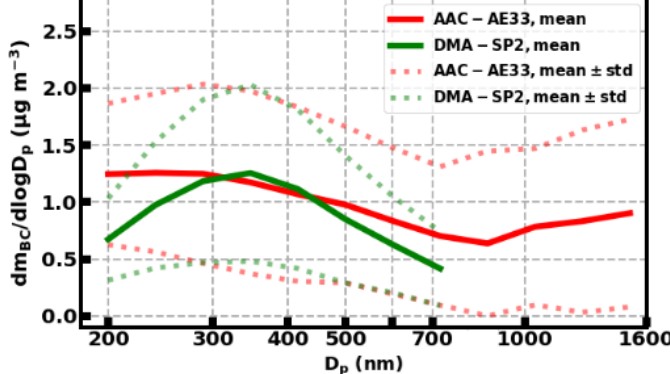


**Figure 3: Mean and std of BCMSD$_{AAC-AE33}$ (green) and BCMSD$_{DMA-SP2}$ (red) during the measurement period. Mean**

**BCMSD was denoted by the solid line. Mean $\pm$ std of BCMSD was denoted by the dotted line.**




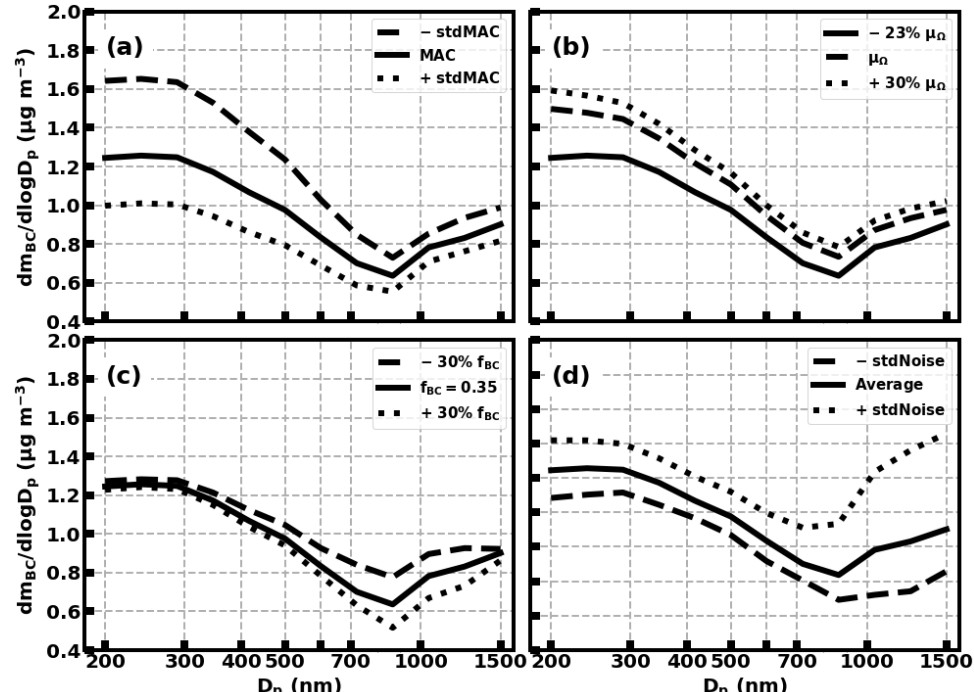

**Figure 4: Uncertainty in $BCMSD_{AAC-AE33}$ arising from (a) MAC lookup table, (b) transfer function inversion, (c) $f_{BC}$ and (d) instrumental noise. The solid lines in each panel were the same and are mean $BCMSD_{AAC-AE33}$ during the measurement period. The dotted lines and dashed lines in each panel were mean $BCMSD_{AAC-AE33} \pm$ standard deviation (std) calculated from (a) MAC + std of MAC and MAC – std of MAC, (b) default $\mu_\Omega$ and 1.3 times default $\mu_\Omega$, (c) $f_{BC}$ of 35% times 0.7 and 1.3, (d) $BCMSD_{AAC-AE33}$ + std of instrumental noise and – std of instrumental noise.**

**Table 1: The Uncertainty in the bulk $m_{BC}$ resulted from MAC lookup table, transfer function inversion, $f_{BC}$ and instrumental noise.**

| Uncertainty source | MAC | Transfer function inversion | $f_{BC}$ | Instrumental noise |
|---|---|---|---|---|
| Uncertainty in $m_{BC,bulk}$ | $-20\% \sim +28\%$ | $-1\% \sim +21\%$ | $-3\% \sim +3\%$ | $-2\% \sim -1\%$ |