# Peer review of "New method to determine equivalent black carbon mass size distribution"

_Atmospheric Measurement Techniques, 2022_

## Author Comment (AC1)

**Response to Anonymous Referee #1**

The presented manuscript describes a new way how to measure size-resolved black carbon mass using a combination of two established aerosol instruments, aerodynamic aerosol classifier and aethalometer. Authors also provide a comparison of the new method to the measurement using SP2 and discuss the uncertainty of the new method. From the presented analyses, the new method seems promising. Response: Thanks for the positive comments.

**Major comments:**

1) The structure of the text may be slightly altered, for example 2. Methods, 2.1. Instrumental setup, 2.2.1. *AE*, 2.2.2. *AAC*, 2.2. Field measurement, 2.3. Size resolved calculations, 2.3.1. binned AC, 2.3.2. BCMSD? Also, the conclusions text is a summary rather than conclusions, and L291 to 296 is a summary of a summary and most of the text could be omitted.

Response: Thanks for your suggestion. And we changed the structure of the text according to your advice. The title of Sect. 5 was changed from "conclusions" to "summary". Text from L291 to L296 was removed.

2) In Fig. 1, and the connected text, some more information on the sampling system would be useful; in the plot, it looks as all the instrumentation was on one inlet. Is it so? What was the cut-off? What was the flow? How did authors solved sampling with instruments with very different flow rates (those could be also added to the picture, for an easier understanding of the sampling)? Was an isokinetic subsampling considered?

Response: Thanks for your comments. We replotted Fig. 1 with more information and added more description of the sampling system in Sect. 2.1. All the instruments used one inlet with cut-off size and flow rate of aerodynamic diameter of 10  $\mu$ m and 16.67 L min-1, respectively. An advanced flow splitter was adopted for isokinetic sampling. There were other instruments, such as nephelometer (Aurora3000, ecotech, Australia, 5 L min-1), also connected to the splitter to make flow rate approximately 16.67 L min-1.

3) The agreement between the new method and SP2 data could be made stronger, for example:

- In Fig. S2, the two regressions are somehow confusing and not explained in the text? Could this be added?
- Could the data from Fig. 2a be presented in a scatter plot with a linear regression to see the agreement?
- Would not be better to correlate the below 720 nm data with those 720 to 1500 nm data to see if these are connected, rather than regress them with data not measured in the size range? (L173 to L177)

Response: Thanks for your comments. We made corresponding changes according to your suggestion. Fig. S2 was replotted into 2 subplots in the supplement and explained in the text. Fig. S2a was the comparison between meBC,bulk,AAC-AE33,200-720 (eBC mass concentration integrated from eBCMSDAAC-AE33 ranging from 200 nm to 720 nm) and mrBC,bulk,DMA-SP2,200-720 (rBC mass concentration integrated from rBCMSDDMA-SP2 ranging from 200 nm to 720 nm) to see the agreement of data from Fig. 2a. Fig. S2b depicted the comparison between meBC,bulk,AAC-AE33,720-1500 (eBC mass concentration integrated from eBCMSDAAC-AE33 ranging from 720 nm to 1500 nm) and mrBC,bulk,DMA-SP2,200-720 to see if these two size ranges were connected.

4) In the uncertainty analysis, if the BCMSD was negative for below 23 % (L227), does it mean it is the limit of the method? If so, this should be somehow more explained and highlighted. If it is not so, what does it mean for the method if it brings negative values...? It seems a larger problem than the resulting uncertainty in Fig. 4b? Similarly, what is the reason of the incomplete parametrization? (L232) Would it be applicable to all measurements? Can the parametrization be improved?

Response: Thanks for your comments. If the eBCMSDAAC-AE33 was negative for  $\mu_{\Omega}$  below 23 %, it did not mean it was the limit of the method. It meant that eBCMSDAAC-AE33 was not valid for  $\mu_{\Omega}$  below 23 % under setting of AAC used in this study (( $Q_{\text{sheath}}, Q_{\text{sample}}$ ) = (7.5 L min-1, 3 L min-1)). eBCMSDAAC-AE33 could be positive for  $\mu_{\Omega}$  below 23 % by increasing  $Q_{\text{sheath}}$ . Because desired  $\mu_{\Omega}$  parameterization was found at 77% of its original value, we did not change  $Q_{\text{sheath}}$  of AAC.

The reason of the incomplete parameterization was that  $\mu_{\Omega}$  was highly dependent on ( $Q_{\text{sheath}}$ ,  $Q_{\text{sample}}$ ) and parametrization scheme proposed by Johnson et al. (2018) ( $\mu_{\Omega,\text{Johnson}}(D_p)$ ) did not consider all possible cases of ( $Q_{\text{sheath}}$ ,  $Q_{\text{sample}}$ ). The parameterization in this study was applicable to measurements where AAC setting was ( $Q_{\text{sheath}}$ ,  $Q_{\text{sample}}$ ) = (7.5 L min-1, 3 L min-1). The parameterization could be improved by studying  $\mu_{\Omega}$  under more cases of ( $Q_{\text{sheath}}$ ,  $Q_{\text{sample}}$ ).

**Technical corrections:**

**1) L101, "a" should be omitted?**

Response: Thanks for your comment. "a" at L101 was deleted in the revised manuscript.

**2) L112, "c" in Cunningham should be in capital?**

Response: Thanks for your comment. "c" in Cunningham at L112 was written in capital in the revised manuscript.

**3) L118, a verb is missing?**

Response: Thanks for your comment. we added a "was" to L118 in the revised manuscript.

**4) L119, why not to be specific and state that AE33 did not need any correction, instead of "the instrument downstream"?**

Response: Thanks for your recommendation. We replaced "data measured by the instrument downstream AAC" with "AE33" in the revised manuscript.

**5) L134, why the constant MAC is stated when not used in the calculations? L134 and 135 may be omitted.**

Response: Thanks for your suggestion. L134 and L135 were removed in the revised manuscript.

**6) L143, one "bin" may be omitted?**

Response: Thanks for your comment. We deleted one "bin" in the revised manuscript.

**7) L164, could be the pollution episodes somehow highlighted in the plot?**

Response: Thanks for your comment. We replotted Fig. 2a and highlighted the pollution episodes with orange shades in the revised manuscript.

**8) In Fig. 3, the dotted line does not denote mean +- std, only std.**

Response: Thanks for your comment. We replotted Fig. 3 and replaced the legend "mean  $\pm$  std" with "std" in the revised manuscript.

**9) L205, S2b should be S3b?**

Response: Thanks for your comment. We changed "S2b" into "S3b" in the revised manuscript.

**10) L230, is the 1 % uncertainty or difference?**

Response: Thanks for your comment. It should be "1% difference", and we replaced "uncertainty" with "difference" in L230 of the revised manuscript.

**11) L253 and 254, what was the correlation between the two lines?**

Response: Thanks for your comment. determination coefficient ( $R^2$ ), slope ( $b_1$ ) and intercept ( $b_0$ ) between the instrumental noise and  $\sigma_{ab,bulk}$  were 0.0, 0.0 and 0.1 Mm-1, respectively. We added the correlation results to the

revised manuscript.

**12) L264, limiting instead of limited?**

Response: Thanks for your comment. It should be "limiting" and we changed "limited" to "limiting" in the revised manuscript.

**References**

Johnson, T. J., Irwin, M., Symonds, J. P. R., Olfert, J. S., and Boies, A. M.: Measuring aerosol size distributions with the aerodynamic aerosol classifier, Aerosol Science and Technology, 52, 655-665, 10.1080/02786826.2018.1440063, 2018.

---

## Author Comment (AC2)

Response to Anonymous Referee #1

*The presented manuscript describes a new way how to measure size-resolved black carbon mass using a combination of two established aerosol instruments, aerodynamic aerosol classifier and aethalometer. Authors also provide a comparison of the new method to the measurement using SP2 and discuss the uncertainty of the new method. From the presented analyses, the new method seems promising.*

Response: Thanks for the positive comments.

**Major comments:**

*1) The structure of the text may be slightly altered, for example 2. Methods, 2.1. Instrumental setup, 2.2.1. AE, 2.2.2. AAC, 2.2. Field measurement, 2.3. Size resolved calculations, 2.3.1. binned AC, 2.3.2. BCMSD? Also, the conclusions text is a summary rather than conclusions, and L291 to 296 is a summary of a summary and most of the text could be omitted.*

Response: Thanks for your suggestion. And we changed the structure of the text according to your advice. The title of Sect. 5 was changed from "conclusions" to "summary". Text from L291 to L296 was removed.

*2) In Fig. 1, and the connected text, some more information on the sampling system would be useful; in the plot, it looks as all the instrumentation was on one inlet. Is it so? What was the cut-off? What was the flow? How did authors solved sampling with instruments with very different flow rates (those could be also added to the picture, for an easier understanding of the sampling)? Was an isokinetic subsampling considered?*

Response: Thanks for your comments. We replotted Fig. 1 with more information and added more description of the sampling system in Sect. 2.1. All the instruments used one inlet with cut-off size and flow rate of aerodynamic diameter of 10 μm and 16.67 L min$^{-1}$, respectively. An advanced flow splitter was adopted for isokinetic sampling. There were other instruments, such as nephelometer (Aurora3000, ecotech, Australia, 5 L min$^{-1}$), also connected to the splitter to make flow rate approximately 16.67 L min$^{-1}$.

*3) The agreement between the new method and SP2 data could be made stronger, for example:*

- *In Fig. S2, the two regressions are somehow confusing and not explained in the text? Could this be added?*

- *Could the data from Fig. 2a be presented in a scatter plot with a linear regression to see the agreement?*

- *Would not be better to correlate the below 720 nm data with those 720 to 1500 nm data to see if these are connected, rather than regress them with data not measured in the size range? (L173 to L177)*

Response: Thanks for your comments. We made corresponding changes according to your suggestion. Fig. S2 was replotted into 2 subplots in the supplement and explained in the text. Fig. S2a was the comparison between $m_{eBC,bulk,AAC-AE33,200-720}$ (eBC mass concentration integrated from eBCMSD$_{AAC-AE33}$ ranging from 200 nm to 720 nm) and $m_{rBC,bulk,DMA-SP2,200-720}$ (rBC mass concentration integrated from rBCMSD$_{DMA-SP2}$ ranging from 200 nm to 720 nm) to see the agreement of data from Fig. 2a. Fig. S2b depicted the comparison between $m_{eBC,bulk,AAC-AE33,720-1500}$ (eBC mass concentration integrated from eBCMSD$_{AAC-AE33}$ ranging from 720 nm to 1500 nm) and $m_{rBC,bulk,DMA-SP2,200-720}$ to see if these two size ranges were connected.

*4) In the uncertainty analysis, if the BCMSD was negative for below 23 % (L227), does it mean it is the limit of the method? If so, this should be somehow more explained and highlighted. If it is not so, what does it mean for the method if it brings negative values…? It seems a larger problem than the resulting uncertainty in Fig. 4b? Similarly, what is the reason of the incomplete parametrization? (L232) Would it be applicable to all measurements? Can the parametrization be improved?*

Response: Thanks for your comments. If the eBCMSD$_{AAC-AE33}$ was negative for $\mu_\Omega$ below 23 %, it did not mean it was the limit of the method. It meant that eBCMSD$_{AAC-AE33}$ was not valid for $\mu_\Omega$ below 23 % under setting of AAC used in this study (($Q_{sheath}$, $Q_{sample}$) = (7.5 L min$^{-1}$, 3 L min$^{-1}$)). eBCMSD$_{AAC-AE33}$ could be positive for $\mu_\Omega$ below 23 % by increasing $Q_{sheath}$. Because desired $\mu_\Omega$ parameterization was found at 77% of its original value, we did not change $Q_{sheath}$ of AAC.

The reason of the incomplete parameterization was that $\mu_\Omega$ was highly dependent on ($Q_{sheath}$, $Q_{sample}$) and parametrization scheme proposed by Johnson et al. (2018) ($\mu_{\Omega,Johnson}(D_p)$) did not consider all possible cases of ($Q_{sheath}$, $Q_{sample}$). The parameterization in this study was applicable to measurements where AAC setting was ($Q_{sheath}$, $Q_{sample}$) = (7.5 L min$^{-1}$, 3 L min$^{-1}$). The parameterization could be improved by studying $\mu_\Omega$ under more cases of ($Q_{sheath}$, $Q_{sample}$).

**Technical corrections:**

*1) L101, "a" should be omitted?*

Response: Thanks for your comment. "a" at L101 was deleted in the revised manuscript.

*2) L112, "c" in Cunningham should be in capital?*

Response: Thanks for your comment. "c" in Cunningham at L112 was written in capital in the revised manuscript.

*3) L118, a verb is missing?*

Response: Thanks for your comment. we added a "was" to L118 in the revised manuscript.

*4) L119, why not to be specific and state that AE33 did not need any correction, instead of "the instrument downstream"?*

Response: Thanks for your recommendation. We replaced "data measured by the instrument downstream AAC" with "AE33" in the revised manuscript.

*5) L134, why the constant MAC is stated when not used in the calculations? L134 and 135 may be omitted.*

Response: Thanks for your suggestion. L134 and L135 were removed in the revised manuscript.

*6) L143, one "bin" may be omitted?*

Response: Thanks for your comment. We deleted one "bin" in the revised manuscript.

*7) L164, could be the pollution episodes somehow highlighted in the plot?*

Response: Thanks for your comment. We replotted Fig. 2a and highlighted the pollution episodes with orange shades in the revised manuscript.

*8) In Fig. 3, the dotted line does not denote mean +- std, only std.*

Response: Thanks for your comment. We replotted Fig. 3 and replaced the legend "mean $\pm$ std" with "std" in the revised manuscript.

*9) L205, S2b should be S3b?*

Response: Thanks for your comment. We changed "S2b" into "S3b" in the revised manuscript.

*10) L230, is the 1 % uncertainty or difference?*

Response: Thanks for your comment. It should be "1% difference", and we replaced "uncertainty" with "difference" in L230 of the revised manuscript.

*11) L253 and 254, what was the correlation between the two lines?*

Response: Thanks for your comment. determination coefficient ($R^2$), slope ($b_1$) and intercept ($b_0$) between the instrumental noise and $\sigma_{ab,bulk}$ were 0.0, 0.0 and 0.1 Mm$^{-1}$, respectively. We added the correlation results to the

revised manuscript.

*12) L264, limiting instead of limited?*

Response: Thanks for your comment. It should be "limiting" and we changed "limited" to "limiting" in the revised manuscript.

**References**

Johnson, T. J., Irwin, M., Symonds, J. P. R., Olfert, J. S., and Boies, A. M.: Measuring aerosol size distributions with the aerodynamic aerosol classifier, Aerosol Science and Technology, 52, 655-665, 10.1080/02786826.2018.1440063, 2018.

---

## Author Comment (AC3)

Response to Anonymous Referee #2

*The paper presents a new method to quantify elemental carbon mass size distribution using size-resolved absorption coefficient measured by an aerodynamic aerosol classifier in tandem with an aethalometer. It aims to demonstrate the feasibility of the method by comparing the measurement results with that of refractory black carbon mass size distribution measured by a differential mobility analyzer in tandem with a single particle soot photometer (DMA – SP2) during a field measurement in Yangtze River Delta. The impact of different assumptions in calculation processes on EC mass size distribution is also presented.*

*The method presented in the paper is of general interest to the scientific community with application in the climate and air quality sectors. A complete understanding of the black carbon cycle and radiative impacts requires knowledge of processes related to the temporal and spatial evolution of BC size distribution. Although a variety of techniques are available to measure BC size distribution, several limitations of these techniques relates to time resolution, instrument cost and complex data analysis.*

Response: Thanks for your comments.

**Major comments:**

*1) The authors should follow the recommendation by Petzold et al. (2013) to properly clarify the terms used for BC. Due to their design the various BC measuring techniques used in the paper are based on different physical and chemical BC properties. This generates difficulties when comparing data from the different techniques because it may provide different BC concentrations. Since there is no recognized technique to quantify BC in the atmosphere, Petzold et al. (2013) established terminology recommendations for reporting BC results. The terms "refractory BC" (rBC) and "equivalent BC" (eBC) should be used to refer to the BC mass concentration quantified using SP2 and aethalometer, respectively. Internally mixed BC particles composed of both BC and non-BC material should be designated as "BC-containing particles".*

Response: Thanks for your recommendation. We agreed with you and the revisions were made according to your suggestion. The revisions improved the description of our study. BC measured by SP2 and aethalometer was termed as equivalent BC (eBC) and refractory BC (rBC), respectively. Internally mixed BC particles composed of both BC and non-BC material were referred to as BC-containing particles.

*2) The significance of the study should be strengthened. The main motivation of the proposed new method is to extend the detection range of BC size distribution up to 1.5 μm, while the widely used SP2 instrument can't measure rBC above 600 nm. The mass median diameter of fresh BC is normally distributed within the range of 30 – 200 nm. If they exist in the atmosphere, I guess superlarge-sized BC*

*may be formed by massive coating or superaggregation. What is the state of the art on these coarse BC particles? Have they already been observed in the atmosphere using other measurement techniques? Are they formed in the atmosphere by massive coating or superaggregation? What is their contribution to the total BC concentration, BC radiative forcing and CCN ability?*

Response: Thanks for your insight comments. Yes, we agreed with you that coarse BC was formed by massive coating or superaggregation. Wang et al. (2022) found that coarse BC was formed by massive coating or superaggregation with transmission electron microscopy. Contribution of coarse BC to total BC concentration could be as large as 50 % (Wang et al., 2022). The contribution of coarse BC to radiative forcing and CCN ability was unclear so far.

**3) There is lack of information on the measurement setup. There is no information on the aerosol sampling (inlet used for aerosol sampling, cut-off diameter of the sampling inlet, aerosol conditioner to dry,…), which need careful consideration for interpreting the results. In addition the authors did not provide any information regarding the calibration of the instruments. Have each instrument been calibrated individually using spherical monodisperse particles? Which detector was calibrated and used to analyze SP2 measurements? The accuracy of the rBC mass concentration measured by SP2 is strongly dependent on the calibration material chosen. This should be mentioned in the paper. Without these information, the reliability of the data is questionable.**

Response: Thanks for your comments. We input more information on the aerosol sampling system in Sect. 2.1 and Fig. 1 was replotted accordingly.

AAC, AE33, APS and SMPS were calibrated by corresponding manufacturers before the field measurement. SP2 was calibrated by monodispersed Aquadag soot particles. Incandescence high gain channel and incandescence low gain channel were calibrated and used to analyze SP2 measurement. The above information was added to Sect. 2.1.3 of the revised manuscript.

**4) There are some methodology and assumptions in calculation processes of rBC mass size distribution measured by the DMA-SP2 that are not discussed. Just as in a typical SMPS data analysis, the rBC mass size distribution need to be calculated from the raw observations of rBC number (or mass) versus voltage using an inversion code. This inversion depends on particle charging efficiencies and the DMA transfer function. The authors should clearly explain the method they used to correct the size distribution measured by the DMA-SP2 to output the information for singly charged particles only. In addition the particle size selected by the DMA is an electrical mobility diameter, which is highly sensitive to the particle shape. If the authors neglected the dependence of rBC asphericity, they shoud mention it in the paper. The authors could**

*also analyze in section 4 the possible uncertainty in deriving rBC diameters from DMA-SP2 owing to the dynamic shape factor they used.*

Response: Thanks for your comments. We agreed with the comments. We explained the method to correct the size distribution measured by DMA – SP2 to output the information for singly charged particles in Sect. 2.1.3 of the revsied manuscript. We also metioned that rBC asphericity was neglected in Sect. 2.1.3.
* * *
**Specific comments:**

*1) L80-85: Why did not you use the SP2 alone to obtain rBC mass size distribution ? Please add the size range of the DMA-SP2.*

Response: Thanks for your comments. The reason why DMA – SP2 was used rather than SP2 alone was that $D_p$ could be directly measured by DMA. If SP2 was used alone, $D_p$ was calculated from Mie theory with assumed inputs (Taylor et al., 2015). We added the size range of DMA – SP2 in Sect. 2.1.3.

*2) L98: Where was C measured in Zhao et al. (2020b)?*

Response: Thanks for your comment. C in our manuscript was "multiple-scattering correction factor ($C_f$)" defined in the section "INTRODUCTION" of Zhao et al. (2020b), specifically in Page 1834. The discussion of $C_f$ could be found in subsection "Multiple-scattering Correction Factor" of section "RESULTS AND DISCUSSION", specifically from Page 1837 to Page 1838. C was changed into $C_f$ in our revised manuscript to consistent with Zhao et al. (2020b).

*3) L205: Fig. S3b instead of S2b.*

Response: Thanks for your comment. Fig. S2b was changed to Fig. S3b in the revised manuscript.

**References**

Petzold, A., Ogren, J. A., Fiebig, M., Laj, P., Li, S. M., Baltensperger, U., Holzer-Popp, T., Kinne, S., Pappalardo, G., Sugimoto, N., Wehrli, C., Wiedensohler, A., and Zhang, X. Y.: Recommendations for reporting "black carbon" measurements, Atmospheric Chemistry and Physics, 13, 8365-8379, 10.5194/acp-13-8365-2013, 2013.

Taylor, J. W., Allan, J. D., Liu, D., Flynn, M., Weber, R., Zhang, X., Lefer, B. L., Grossberg, N., Flynn, J., and Coe, H.: Assessment of the sensitivity of core / shell parameters derived using the single-particle soot photometer to density and refractive index, Atmos. Meas. Tech., 8, 1701-1718, 10.5194/amt-8-1701-2015,

2015.

Wang, J. D., Wang, S. X., Wang, J. P., Hua, Y., Liu, C., Cai, J., Xu, Q. C., Xu, X. T., Jiang, S. Y., Zheng, G. J., Jiang, J. K., Cai, R. L., Zhou, W., Chen, G. Z., Jin, Y. Z., Zhang, Q., and Hao, J. M.: Significant Contribution of Coarse Black Carbon Particles to Light Absorption in North China Plain, Environ. Sci. Technol. Lett., 9, 134-139, 10.1021/acs.estlett.1c00953, 2022.

---

## Author Response (AR2)

Response to Editor

*Please apologize for the delay in editing this paper. I believe the paper was greatly improved after you took into account comment from both reviewers.*

Response: Thanks for the positive comment.

*It is not necessary to go through another round of external reviews, but I would like to subject publishing to a number of small revisions, listed below:*

*1) I suggest to keep Black Carbon in title and not equivalent Black Carbon, as it can be used as a generic term in this case (in agreement with Petzold et al., recommendation). In the text, please systematically use eBC (avoiding to switch from equivalent-BC to eBC) as recommended in Petzold et al..*

Response: Thanks for your suggestion. We changed the "equivalent Black Carbon" into "Black Carbon" in the title and systematically used eBC in the text according to your advice.

*2) In Experimental methods, I would keep the section titles with the full name, not abbrevations (or full name (abbreviation)).*

Response: Thanks for your comment. The section titles were replaced with full name.

*3) P4, line 103, I believe the comment from Rev. was referring to the fact that referring to the place where Zhao et al. performed their measurements. This is important that measurements were performed at the same place, considering geographical variability of BC properties. In addition, please check exact reference for Zhao et al., 2020b.*

Response: Thanks for your comments. The Zhao et al., 2020b performed their measurement in Beijing and Taizhou of China, respectively. Beijing and Taizhou were 890 km part and the scattering parameter $C_f$ was measured to be 2.9 in both Beijing and Taizhou. The measurement site in this study (Changzhou) was close to (70 km away from) Taizhou. The pollution condition in Changzhou was similar to that in Taizhou. Both Changzhou and Taizhou were in the Yangtze River Delta and between two megacities, namely Nanjing and Shanghai. Therefore, $C_f$ was set 2.9 in this study.

*4) Please, provide a check for English in section 2.1.3. Some sentences might be improved.*

Response: Thanks for your comment. English in section 2.1.3 was improved in the revised manuscript.

*5) P7, line 209, I am wondering if the term "exclusively" would rather be "independently" or "separately".*

Response: Thanks for your suggestion. "exclusively" was changed into "independently" in the revised

manuscript.

*6) P7, line 211, I am wondering if the use of past tense is appropriate. I have the impression that the sentence should be rather "Therefore, it IS highly necessary to measure BCMSD with wider Dp range for better estimation of $m_{BC,bulk}$ and deeper understanding of BC evolution in the atmosphere", indicating a more general statement than a step in your scientific analysis.*

Response: Thanks for your suggestion. We changed "was" into "is".

*7) P10, line 310, an overall conclusion on uncertainty could be added, then used again in the conclusions. In it important that, at the end, a reader can rapidly evaluate how performant is, considering all uncertainties, your technique for retrieving size-distribution. I am not surprised that MAC is the higher source of uncertainty but was expecting higher values. At one single place, MAC variability is quite important (see Zanatta et al. (2018)).*

Response: Thanks for your comments. An overall conclusion on uncertainty was add in line 310.

As shown in the study by Zanatta et al. (2018), the lensing effect could lead to 54% increase of MAC and Fig. 7 in Zanatta et al. (2018) clearly illustrated the large variability of MAC. The reason why the uncertainty from MAC was lower than expected might be that one of the various factors influencing variability of MAC, namely lensing effect, was considered in our study by adopting method developed by Zhao et al. (2021).

*8) P10, line 312, again, use of past tense does not seem appropriate here. Perhaps, "not sufficient" is better than "very limiting".*

Response: Thanks for your suggestion. The past tense was changed to present tense and "very limiting" was replaced with "not sufficient".

**References**

Zanatta, M., Laj, P., Gysel, M., Baltensperger, U., Vratolis, S., Eleftheriadis, K., Kondo, Y., Dubuisson, P., Winiarek, V., Kazadzis, S., Tunved, P., and Jacobi, H. W.: Effects of mixing state on optical and radiative properties of black carbon in the European Arctic, Atmospheric Chemistry and Physics, 18, 14037-14057, 10.5194/acp-18-14037-2018, 2018.

Zhao, W. L., Tan, W. S., Zhao, G., Shen, C. Y., Yu, Y. L., and Zhao, C. S.: Determination of equivalent black carbon mass concentration from aerosol light absorption using variable mass absorption cross section, Atmospheric Measurement Techniques, 14, 1319-1331, 10.5194/amt-14-1319-2021, 2021.